# Factors Related to Oral Problems in Patients with Prolonged Disorders of Consciousness in Long-Term Care: A Cross-Sectional Study

**DOI:** 10.3390/healthcare11111622

**Published:** 2023-06-01

**Authors:** Kanako Yoshimi, Kazuharu Nakagawa, Kohei Yamaguchi, Ayako Nakane, Masaharu Hayashi, Rie Miyata, Yumi Chiba, Haruka Tohara

**Affiliations:** 1Department of Dysphagia Rehabilitation, Division of Gerontology and Gerodontology, Graduate School of Medical and Dental Sciences, Tokyo Medical and Dental University, Tokyo 113-8510, Japan; k.yoshimi.gerd@tmd.ac.jp (K.Y.); k.yamaguchi.swal@tmd.ac.jp (K.Y.); a.nakane.swal@tmd.ac.jp (A.N.); h.tohara.swal@tmd.ac.jp (H.T.); 2School of Nursing, College of Nursing and Nutrition, Shukutoku University, Chiba 260-8703, Japan; hayashi-ms@igakuken.or.jp; 3Department of Pediatrics, Tokyo-Kita Medical Center, Tokyo 115-0053, Japan; rie88miyata@yahoo.co.jp; 4Cancer/Advanced Adult Nursing, Department of Nursing, Graduate School of Medicine, Yokohama City University, Yokohama 236-0004, Japan; ychiba@yokohama-cu.ac.jp

**Keywords:** disorders of consciousness, oral function, oral intake, long-term care

## Abstract

Factors influencing oral problems, such as malocclusion and oral motor dysfunction, in patients with prolonged disorders of consciousness (DOC) remain unclear. This study aimed to clarify the relationship between oral problems and physical function, communication, respiration, and oral intake status, as well as related factors in patients with DOC receiving long-term care at home. A cross-sectional study was conducted in October 2018; 127 patients who developed DOC > 5 years ago were analyzed. The differences between patients with and without oral problems were examined, and a binomial logistic regression analysis was performed to examine factors associated with oral problems, with the presence of oral problems as the dependent variable, and age, the number of years since onset, drooling, oral intake status, and the presence of a family dentist as explanatory variables. A post hoc power analysis of the binomial logistic regression analysis for oral problems (odds ratio: 2.05, alpha value: 0.05, incidence of oral problems: 0.80, and total sample size: 127) demonstrated an observed power of 93.09%. Oral intake status (*p* = 0.010) and the number of years since onset (*p* = 0.046) were significantly related to oral problems. Preventive oral management and rehabilitation from the early stage after onset may be effective for oral problems in patients with DOC.

## 1. Introduction

The oral cavity is composed of various muscles as well as soft and hard tissues. Tooth position and stable occlusion are maintained by intrinsic forces from soft tissues, such as the lips and tongue, extrinsic forces from habits and orthodontic treatment, occlusal forces, and periodontal tissues [1]. For example, habits such as mouth breathing and thumb sucking in childhood can cause malocclusion [2,3]. Overstrain and decreased muscle strength and motor function of the perioral muscles may affect the orofacial region. Additionally, malocclusion may lead to inflammation and abnormalities of the periodontal tissues [4]. Furthermore, impaired tongue movement because of significant malocclusion may cause swallowing disorders and respiratory problems owing to glossoptosis [5]. These oral problems are classified as organic or functional problems, and they interact with each other in a complex manner and influence various aspects, such as oral function, swallowing function, and respiratory status. Most previous studies on oral problems have focused on patients in developmental stages or those with congenital diseases [6,7,8]. However, only a few clinical reports have been published on dental problems and treatment in patients with acquired disabilities. The causes of acquired disability, including disease and injury, vary along with their severity.

A disorder of consciousness (DOC) is a long-term condition characterized by difficulty in independent ambulation, following directions, and verbal communication due to cerebral lesions [9,10]. DOC includes the unresponsive wakefulness syndrome (UWS) and “minimally conscious state” (MCS). UWS is defined as a state wherein sleep–wakefulness is maintained but no behavioral evidence of awareness of self or environment is observed, whereas MCS is defined as a state wherein patients have some extent of awareness of themselves and the surrounding environment. The number of patients with DOC receiving home care is increasing due to advances in medical care, necessitating further care and rehabilitation of these patients. Oral problems, such as malocclusion, mouth opening and closing disorders, and decreased tongue motor function, are observed in many patients with prolonged DOC [5]. However, the mechanisms underlying these problems are not well understood. A dental treatment protocol has not been established for these patients, and the factors that affect oral problems and their relationship with general conditions are unknown. Therefore, this study aimed to clarify the relationship between oral problems and physical function, communication, respiration, and oral intake status, as well as the related factors in patients with prolonged DOC receiving long-term care at home.

## 2. Materials and Methods

A cross-sectional study using a questionnaire was conducted. Cooperation was obtained from the Japanese Association of the Families of Patients with Persistent Consciousness Disorder and its supporting organizations. The included patients and their family members belonged to these organizations. Their family members completed the questionnaire. A total of 508 questionnaires were sent by mail to the members in October 2018, and anonymized answers were obtained by the end of November 2018. The study protocol was approved by the Institutional Review Board of Tokyo Medical and Dental University (approval ID: D2018-13). We explained the study aims and obtained written informed consent from the family members of the patients. This was a human observational study, the manuscript conforms to the STROBE guidelines, and the survey was conducted ethically in accordance with the World Medical Association Declaration of Helsinki.

### 2.1. Data Collection

As no previous studies have investigated the oral status of patients receiving home care, the questionnaire was originally designed for this study. Basic information about the participants and items that may be relevant to the oral region and swallowing function were established. For the self-reporting of oral problems, several oral pictures and descriptions were shown, and the family members were asked to identify the picture that they thought most represented the participants’ oral status (Figure 1).

The survey items were as follows:Age, sex, the number of years since the onset of the disorder, and the cause of onset.Physical function: walking with assistance, sitting (end sitting position), sitting (with backrest), can only roll over, and total assistance.Communication disorder: communicating with words, only speaking words, only emotional expressions, and no emotional expressions.Respiratory care: none, oxygen administration, simple tracheostomy, or laryngo-tracheal separation.Drooling and airway suctioning: none, rarely, sometimes, and every day.Food intake: total oral diet, tube-dependent with oral intake of food or liquid, and nothing by mouth.Oral problems: malocclusion, inability to open or close the mouth, dislocation of the temporomandibular joint, and glossoptosis. Multiple answers were considered acceptable. Excludes problems, such as dental caries, periodontal disease, and mucosal diseases.Presence of a family dentist: none, a dental office, house call dentistry, a university hospital, and others.

### 2.2. Data Analysis

Since the outcome in this study was defined as oral problems, including malocclusion of the dentition, the analysis included those with long-term care for more than 5 years, considering the time course of the problem. First, we compared the effects of the presence of oral problems. On the basis of the answers to the questionnaire, the patients were classified as those with and without oral problems, and the Mann–Whitney U test and chi-square test were used to examine the differences in each item between the two groups. Next, a binomial logistic regression analysis was performed to examine factors associated with oral problems, with the presence of oral problems serving as the dependent variable, and age, the number of years since onset, drooling, oral intake status, and presence of a family dentist serving as explanatory variables. All statistical analyses were performed using SPSS 25.0 (IBM Inc., Tokyo, Japan), and the significance level was set at *p* < 5%.

## 3. Results

Two hundred and fifty-seven questionnaires were returned, resulting in a response rate of 50.6%. Fifty-seven participants with missing data were excluded from the study. Furthermore, 73 participants with less than 5 years since the DOC onset were excluded. In total, 25 participants had less than 1 year of onset, while 23, 8, and 17 participants had an onset of 2, 3, and 4 years, respectively. The analysis was conducted among 127 participants with 5 to 32 years after the DOC onset.

The post hoc power analysis of the binomial logistic regression analysis for oral problems (odds ratio [OR]: 2.05, alpha value: 0.05, incidence of oral problems: 0.80, and total sample size: 127) demonstrated an observed power of 93.09%.

### 3.1. Participant Characteristics

The characteristics of the participants are shown in Table 1. The study population included 82 men and 45 women, and the median age of the participants was 60 years (30–81 years). The median number of years since the DOC onset was 12 years (5–32 years). The causes of DOC were traffic injuries in 67 participants, cerebrovascular disease in 20, hypotonic ischemic encephalopathy in 16, medical accidents in 8, cardiovascular disease in 6, other diseases in 2, industrial accidents in 1, and others in 7.

Regarding physical function, 47% of the participants had difficulty moving independently. Regarding communication disorders, 61% had only emotional expression, 23% had no emotional expression, and 84% had difficulty communicating by speech. Regarding respiratory care, approximately half of the participants had undergone a tracheostomy. Drooling was observed in 80% of the participants, although the frequency of drooling varied. The frequency of airway suctioning varied. Regarding oral intake status, half of the participants were on tube feeding only, and 37% were on a combination of oral intake and tube feeding. Approximately 90% of the participants were on tube feeding. Oral problems were “none” (20%) and 80% of all the participants had some type of oral problem in any of the other five categories; 57% had “malocclusion”. Regarding family dentistry, domiciliary dentistry was the most common (approximately 60%; Table 1).

### 3.2. Comparisons Based on the Presence of Oral Problems

A significant difference between the two groups in terms of “drooling” (*p* = 0.031), “airway suctioning” (*p* = 0.049), and the “oral intake status” (*p* = 0.043) (Table 2) was observed.

### 3.3. Factors Affecting Oral Problems

The binary logistic regression analysis showed that oral problems were associated with “years since onset” (OR: 1.50, 95% confidence interval [CI]: 0.99–1.22, *p* = 0.046) and “oral intake status” (OR: 2.05, 95% CI: 1.16–2.96, *p* = 0.010). Oral problems were found in approximately 80% of all patients with prolonged DOC and indicated a possible dependence on the number of years since onset and on oral intake. However, the presence of a family dentist was not associated with any oral problems (Table 3).

## 4. Discussion

This is the first study investigating the oral status of patients with prolonged DOC receiving long-term care in Japan. Approximately 80% of the participants had oral problems, with malocclusion being the most predominant. Oral problems were also significantly associated with the frequency of drooling and airway suctioning and with the absence of oral intake. Furthermore, the number of years since onset and oral intake status affected the presence of oral problems.

### 4.1. Relationship between Oral Function and Malocclusion

Malpositioning of the mandibular anterior teeth is directly related to a decrease in the maximum tongue pressure [11]. This finding may be attributed to the posterior positioning of the tongue decreasing the molding effect on the anterior dentition. The individuals participating in this study were adults, and the development of their dentition had already been completed. However, many of these participants were bedridden and likely to keep their mouths open. The tongue may not be in contact with the mandibular anterior teeth when the tongue root sinks under the influence of gravity and the tongue movement in the anterior–posterior direction is reduced due to decreased tongue function; this may lead to tooth inclination and crowding. These findings emphasize the importance of maintaining the function, movement, and proper positioning of the tongue for the prevention of tooth movement. The position of the teeth is determined by the balance between soft tissues, such as the cheeks, lips, and tongue, and can be moved by applying an external force as small as 1.68 g [12,13]. Thus, daily application of weak external forces to the perioral region, including those from supine or lateral lying postures, sustained mouth opening, and excessive tension and relaxation of the perioral region may result in changes in the dentition over time.

The number of years since disease onset was found to be significantly associated with oral problems. The identification of slight changes in the oral status of patients during long-term care is difficult, and these changes are more likely to present as oral problems, such as dental malposition and malocclusion, after the movement of the teeth. Therefore, the prediction of possible future oral problems based on the patient’s oral function and general condition and providing appropriate preventive measures is essential.

In addition, since this study included patients who were in long-term care, their oral status at the onset of the disease was unknown. The baseline oral status at the onset of DOC, such as the presence of malocclusion, oral habits, and oral hypofunction, could affect the process in subsequent years. Screening the oral status at onset and longitudinal follow-up may provide a more specific mechanism for the progression of oral problems.

### 4.2. Relationship between Swallowing Function and Oral Problems

In this study, the number of participants using tube feeding only was significantly higher in the group with oral problems. A previous study investigating the swallowing function of hospitalized patients with DOC from the acute to the convalescent phase evaluated 92 patients using swallowing endoscopy and subsequently compared their fluid intake status. Hypertonicity of the perioral muscles and the absence of oral movements, such as tongue and mastication-like movements, were observed in almost all patients [14], suggesting that patients with DOC are likely to have impaired oral motor function, which makes the transport of food and saliva from the oral cavity to the pharynx difficult.

The participants in this study were long-term care patients with chronic DOC. Therefore, the extent of brain damage and severity of consciousness impairment affected the swallowing function and oral intake status of the patients. However, although passive and reflexive, oral and swallowing movements are routinely performed when patients consume even a small amount of food orally. More than 30 muscles and nerves are coordinated and active during feeding and swallowing [15]. Thus, the stimulation of the perioral and swallowing-related muscles may help prevent oral problems and maintain swallowing function.

Drooling and airway suctioning frequency were significantly high in patients with oral problems. A study on drooling in pediatric patients with cerebral palsy reported that the volume of saliva itself did not increase in these patients; however, the swallowing frequency tended to be low [16]. Saliva cannot be transported from the oral cavity to the pharynx in patients with DOC and impaired oral motor function, resulting in the retention of saliva in the oral cavity and drooling. However, the binomial logistic regression analysis showed no significant difference in drooling, indicating that saliva processing and oral intake should be considered separately. Patients with an impaired oral stage can consume food orally using compensatory approaches, such as inserting a spoon into the posterior region of the tongue or adjusting the posture during eating, in addition to adjusting the diet form [17]. Early dysphagia rehabilitation according to the patient’s condition may also be effective in maintaining oral and pharyngeal functions.

### 4.3. Dental Approach to Oral Problems

This study showed no relationship between the presence of oral problems and the presence of a family dentist. No previous study has identified oral problems specific to patients receiving in-home care or those with prolonged DOC. Therefore, no guidelines or specific measures have been provided in this regard. Nevertheless, dentists and dental hygienists should be aware of the factors underlying oral problems in such patients and perform early preventive oral management in addition to oral hygiene management and dental treatment. A previous study on children has reported the effectiveness of oral myofunctional therapy for the treatment of anterior open bite and tongue dysfunction [18]. Therefore, if spontaneous oral movement is difficult, oral dysfunction and oral problems could be reduced through a combination of oral care and training, such as massaging the perioral muscles and using a hyperactive approach. In addition, since the time spent away from bed is related to swallowing in patients requiring care, a systemic approach that includes the oral region could also be effective [19]. Furthermore, a swallowing assessment should be performed to evaluate the possibility of oral intake at each stage of treatment. Thus, evaluation of the swallowing function, by domiciliary medical and dental teams, of patients receiving home care would be helpful.

A previous study reported that an oral appliance effectively reduced snoring and improved apnea caused by glossoptosis in patients with DOC [20]. If oral problems are likely to develop because of myotonia, hypomyotonia, or a constricted dental arch, early intervention with an oral appliance may help prevent malocclusion and the reduction of tongue space associated with malocclusion as well as benefit respiratory management. Oral problems become difficult to manage once they have already progressed to an extent where the patient or caregiver becomes aware of them. Therefore, establishing evidence for the preventive management of oral problems is essential.

### 4.4. Limitations

This study has some limitations. First, this was a questionnaire-based, fact-finding survey, and the presence of oral problems was reported by the family members. The questionnaire was created by the authors, and bias may have occurred in the survey of family members. Therefore, it is necessary to clarify the mechanisms and factors affecting oral problems from professional and academic perspectives. Second, evaluating the patient’s general condition and swallowing function plays an essential role in investigating the factors influencing oral problems in detail. Oral and swallowing functions may be related to the patient’s general condition and whether swallowing rehabilitation is performed in daily life. Therefore, evaluating the patients’ medical treatment environments is essential. Third, this study did not compare the situation of patients with DOC in Japan with that of those in other countries. Thus, consideration of dental approaches related to in-home care for patients in other countries is required.

## 5. Conclusions

In conclusion, significantly associated factors for oral problems in patients with prolonged DOC in long-term care were oral intake status and the number of years since onset. To prevent oral problems, oral management and rehabilitation should be performed from the early stage after onset, and preventive management for malocclusion may also be effective. These findings also indicate the need to establish a new domiciliary dental approach.

## Figures and Tables

**Figure 1 healthcare-11-01622-f001:**
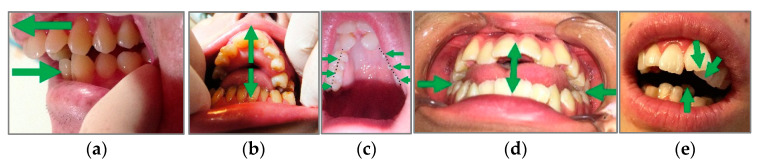
Several oral pictures and descriptions for the self-reporting of oral problems: (**a**) the upper teeth are protruding forward; (**b**) lower teeth are tilted inward; (**c**) posterior teeth are tilted inward; (**d**) the posterior teeth are in occlusion, while the anterior teeth have a gap; and (**e**) teeth are more inclined than before.

**Table 1 healthcare-11-01622-t001:** Patient characteristics (N = 127).

		N (%)
Age, median (min.–max.)		60 (30–81)
Number of years since onset, median (min.–max.)		12 (5–32)
Sex	Male	82 (65)
	Female	45 (35)
Cause of onset	Traffic injuries	67 (52.8)
	Cerebrovascular disease	20 (15.7)
	Hypotonic ischemic encephalopathy	16 (12.6)
	Medical accidents	8 (6.3)
	Cardiovascular disease	6 (4.7)
	Other diseases	2 (1.6)
	Industrial accidents	1 (0.8)
	Others	7 (5.5)
Physical function	Walking with help	6 (5)
	Sitting position	54 (43)
	Can only roll over	7 (5)
	Total assistance	61 (47)
Communication disorder	Communication with words	10 (8)
	Only speak words	10 (8)
	Only emotional expression	77 (61)
	No emotional expression	28 (23)
Respiratory care	None	39 (32)
	Oxygen administration	10 (8)
	Tracheostomy	65 (53)
	Laryngo-tracheal separation	8 (7)
Drooling	None	26 (21)
	Rarely	23 (19)
	Sometimes	34 (28)
	Every day	40 (32)
Airway suctioning	None	32 (32)
	Less than 10 times a day	34 (33)
	More than 10 times a day	35 (35)
Oral intake status	Total oral diet	15 (12)
	Tube-dependent with oral intake of food or liquid	46 (37)
	Nothing by mouth	64 (51)
Oral problems (Multiple answers allowed)	None	25 (20)
	Malocclusion	72 (57)
	Unable to open and close the mouth	37 (28)
	Dislocation of TMJ	7 (6)
	Glossoptosis	24 (18)
	Others	14 (11)
Presence of a family dentist	None	24 (20)
	A dental office	7 (6)
	Domiciliary dentistry	71 (59)
	A university hospital	9 (7)
	Others	10 (8)

TMJ; temporomandibular joint. The questionnaires were answered by the family members of the participants between October 2018 and November 2018 in Japan.

**Table 2 healthcare-11-01622-t002:** Comparison of the presence of oral problems.

		Oral Problems		*p*-Value
		Absent (N = 25)	Present (N = 102)	
Age	Median (min.-max.)	59 (30–76)	63 (30–81)	0.345 ^†^
Number of years since onset	Median (min.-max.)	11 (5–18)	13 (5–32)	0.751 ^†^
Sex	Male	17	65	0.922 ^‡^
	Female	9	36	
Physical function	Total assistance	12	49	0.849 ^‡^
Communication disorder	Only facial expression	17	60	0.578 ^‡^
Respiratory care	Tracheotomy	16	49	0.768 ^‡^
Drooling	None	10	16	0.031 ^‡^*
Airway suctioning	More than 10 times a day	3	32	0.049 ^‡^*
Oral intake status	Nothing by mouth: tube feeding only	9	55	0.043 ^‡^*
Presence of a family dentist	Presence	18	69	0.142 ^‡^

^†^ Mann–Whitney U test, ^‡^ chi-square test, * *p* < 0.05. The questionnaires were answered by the family members of the participants between October 2018 and November 2018 in Japan.

**Table 3 healthcare-11-01622-t003:** Factors related to the oral problem.

	Odds Ratio Exp(β)	95% CI		*p*-Value
Age	0.98	0.94	1.02	0.314
Number of years since onset	1.50	0.99	1.22	0.046 *
Drooling	1.11	0.72	1.74	0.626
Oral intake status	2.05	1.16	2.96	0.010 *
Presence of a family dentist	1.00	0.65	1.56	0.993

Binomial logistic regression analysis, * *p* < 0.05. CI, confidence interval.

## Data Availability

The datasets generated and analyzed during the current study are not public because they contain information that could compromise the privacy of the participants but are available from the corresponding author upon reasonable request.

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
