# Peer review of "Factors Related to Oral Problems in Patients with Prolonged Disorders of Consciousness in Long-Term Care: A Cross-Sectional Study"

_healthcare, 2023, doi:10.3390/healthcare11111622_

Round 1

Reviewer 1 Report

Factors related to oral problems in patients with prolonged disorders of consciousness in long-term care:  a cross-sectional study

Thank you for opportunity to review your novice manuscript. I found it interesting because limited research is available on oral problems in patients with prolonged disorders of consciousness. There are some issues I recommend the authors to improve and clarify in their revision of their manuscript. Like the reason for excluding respondents from certain group with DOC from the analysis and be precise on who the respondents are in the study. I will comment on general and specific parts of the manuscript and the corresponding lines as necessary.

AUTHORS INFORMATION:

Lines 9, 10: Please add the missing spaces after the email addresses.

ABSTRACT:

It is not clear what types of “oral problems”, or symptoms are expected to be present among the subject in the study, until line 96 in the Data collection chapter: (Oral problems: malocclusion, inability to open or close the mouth, dislocation of the temporomandibular joint, and glossoptosis) are set as the dependent variable. Line 27, please be concise when writing P- values/alfa values in your manuscript. In the abstract the p value is presented with capital letter/upper–case P and is non-Italicised, but in line 110 and forward, the p value is italicised. Further in line 145 the p- value is written with a lower–case letter p.

INTRODUCTION:

Lines 56-57: “… DOC receiving at home … is increasing due to…”  the word CARE is missing in this sentence.

MATERIAL AND METHODS:

It is common that health care professions, family or caretakers answer questionnaires for the behalf of their patient, family member and that the patients answer surveys themselves. These groups may answer questionnaire quite differently, based on their understanding, assessment, and experience. Therefore, please note that Line 70: “Their family members completed the questionnaire” and Line 245: “reported by the family members” is inconsistent in the manuscript with the information given in Lines 128 and 146: “The questionnaires were answered by the participants or their caregivers”. It must be absolutely clear who the respondents are in the study and the results presented accordingly.

DATA COLLECTION:

Lines 82-84: I recommend rephrasing these sentences, so it is clear who is answering the questionnaire or self-reporting (DOC patients, family members, or both).

I assume that the “oral pictures” shown, and descriptions were part of the questionnaire. This could be mentioned earlier when introducing the questionnaire for the first time, or in the material and method section as part of the instrument (questionnaire). Definition of malocclusion is missing, and it is unknown how the authors categorised people with malocclusion or not.

Lines 86 -100: The list would be easier to read if the lines were set with hanging index. The number comes first, and the second line does not exceed the numbered line.

Of 508 surveys, 257 were returned, 57 were excluded due to missing data. Lines 113-114: Please justify why information 73 questionnaires (DOC < 5 years form onset) was excluded from the analysis, dropping the sample size from 200 to 127. There should be logical and ethical explanation to this exclusion.

RESULTS:

Table 1, the defined “oral problem” in the data collection section is called “Orofacial problem” in the table, I recommend keeping the text consistent and use either throughout the manuscript.

Line 128/146 Who answered the questionaries the participants (DOC patient or caregivers)?

Lines 137-139: Regarding this sentence: “oral problems were “none” (20%) and “malocclusion” (57%), and 80% of the patients had some sort of oral problem.” It is not possible to see how the authors result that 80% of the sample have “some sort of oral problem” (dependent variable) in Table 1. The “oral problems/orofacial problems” variable is a multiple respond type questions shown in Table 1 with six respond categories. Since, in lines 96-97 the dependent variable “oral problem” has four categorises the respondent must score at least one or more of these four categories to be defined with “oral problems”. The ratio showing (80%) with oral problems is not understandable until Table 2 has been read, showing that 102 out of 127 were classified with “oral problems”.  This must be addressed in the manuscript. Further, in Table 2 it can be seen that “years since onset” is not significantly different between those with and without oral problems. Lines 151-152 referring to factors affecting oral problems using binary logistic regression state: “Oral problems were found in approximately 80% of all patients with prolonged DOC and indicated a possible dependence on the number of years since onset and on oral intake.” How were “number of years since onset” categorised in the regression. This is related to the exclusion of 73 questionnaires from DOC subject with less than five years from onset. If they had been included in the analysis did the data show significant difference at all.

DISCUSSION

Your discussion is sensible and relevant, and I agree that it is necessary to establish a dental approach for patients suffering from DOC. This may include raise awareness of the potential oral problems affecting this group. Therefore, I like to comment on chapter 4.1 Relationship between oral function and malocclusion:

The most frequent oral problem reported in this study was malocclusion, but the authors definition of malocclusion is not mentioned in the manuscript, or when DOC patient was classified with malocclusion for example normal bite, underbite, overbite or any other classification like occluding pairs of teeth etc. These individuals were 30-81 years of age and the onset of DOC ranged from 5 to 32 years. The clinical oral health status of subjects at onset is unknown. Some may have had malocclusion before prolonged DOC or oral dysfunction related to oral habits, chewing, swallowing, missing teeth etc. These factors may affect the severity of malocclusion after the onset of DOC. I agree that it is important to know the risk of oral problems after onset of DOC, but studies that shows progression of oral problems in this group is needed to establish this knowledge. Baseline oral health examination and longitudinal follow up might show the relevant problems more accurately. These facts might be addressed in your manuscript for the reader.

REFERECES

Please note some of your references are reporting developmental issues related to malocclusion among children (lines 288, 298, 314) but this cross-sectional study is conducted among 30–80 years old subjects. I kindly ask, if these references should not rather be used to describe similar age groups rather than adults. 

Line 305: Please correct misspelling of buccinator.

Reviewer 2 Report

An interesting and well-written piece. 

Line 68: “Association of the Families of Patients with Persistent Consciousness Disorder” I would make comment on which country this relates to. 

Line 126: a formatting issue with the table- would be better to more clearly delineate between the different questions 

132: how many is “many”? Give a percentage. 

139: “domiciliary” is generally the term used instead of “home-care 

143: it is not clear to me what is meant by “suctioning”. In a dental setting, I would consider this to be the use of a saliva ejector or similar. Please explain what you mean by this. 

In Table 1, you refer to “orofacial problem”. However, on line 137, you refer to “oral problems". Are these referring to the same thing? If so, keep consistent. 

If you are not discussing what could be considered oral problems such as dental caries, candidiasis etc., you should make note of that. You may need to change the title to something along the lines of “oral function problems, instead of just “oral problems”.
